# Characterization of Extreme Wave Conditions for Wave Energy Converter Design and Project Risk Assessment

**Vincent S. Neary [1],\*, Seongho Ahn [1], Bibiana E. Seng [2], Mohammad Nabi Allahdadi [3], Taiping Wang [4], Zhaoqing Yang [4] and Ruoying He [3]**

[1]   Sandia National Laboratories, Water Power Technologies, Albuquerque, NM 87185, USA
[2]   Department of Statistics, The Pennsylvania State University, State College, PA 16802, USA
[3]   Department of Marine, Earth, and Atmospheric Sciences, North Carolina State University, Raleigh, NA 27606, USA
[4]   Pacific Northwest National Laboratory, Seattle, WA 98109, USA
\*   Correspondence: vsneary@sandia.gov

**Abstract:** Best practices and international standards for determining *n*-year return period extreme wave (sea states) conditions allow wave energy converter designers and project developers the option to apply simple univariate or more complex bivariate extreme value analysis methods. The present study compares extreme sea state estimates derived from univariate and bivariate methods and investigates the performance of spectral wave models for predicting extreme sea states at buoy locations within several regional wave climates along the US East and West Coasts. Two common third-generation spectral wave models are evaluated, a WAVEWATCH III®model with a grid resolution of 4 arc-minutes (6–7 km), and a Simulating WAves Nearshore model, with a coastal resolution of 200–300 m. Both models are used to generate multi-year hindcasts, from which extreme sea state statistics used for wave conditions characterization can be derived and compared to those based on in-situ observations at National Data Buoy Center stations. Comparison of results using different univariate and bivariate methods from the same data source indicates reasonable agreement on average. Discrepancies are predominantly random. Large discrepancies are common and increase with return period. There is a systematic underbias for extreme significant wave heights derived from model hindcasts compared to those derived from buoy measurements. This underbias is dependent on model spatial resolution. However, simple linear corrections can effectively compensate for this bias. A similar approach is not possible for correcting model-derived environmental contours, but other methods, e.g., machine learning, should be explored.

**Keywords:** extreme significant wave height; wave hindcast; wave energy resource assessment; WEC design

## 1. Introduction

Best practices for determining the site-specific environmental conditions, design load cases (DLC), and load responses for maritime structures and their subsystems (e.g., offshore oil platforms, offshore floating wind turbines, wave energy converters, and mooring systems) are found in a variety of international standards, e.g., [1–3]. Some of the most common load types experienced by maritime structures include hydrostatic, hydrodynamic (currents, waves), and aerodynamic (wind) loads. Design conditions include transport, operation under normal conditions occurring every year, and survival under extreme or abnormal environmental conditions for a range of return periods (e.g., 1, 5, 50,

100-years) [4]. As specified by these international standards and guidelines, DLCs are to be constructed for each design/environmental condition and different combinations of load types.

Extreme wave statistics, like the 50-year significant wave height, $H_{s(50)}$, or 50-year sea states, $(H_s, T_e)_{50}$, are key metrics used to characterize extreme wave conditions for a host of ocean (offshore) and coastal engineering applications [4], including offshore wind and marine energy. In marine energy, the significant wave height recurring every fifty years on average, $H_{s(50)}$, has also been proposed as an indicator of project risk using a relative risk ratio, $H_{s(50)}/H_{s(mean)}$ [5,6]. In addition, the International Electrotechnical Commission (IEC) recommends a design standard for wave energy converters (WEC) that requires $H_{s(50)}$ for building extreme condition DLCs for a WEC in a parked-survival condition, and $H_{s(1)}$ for building abnormal condition DLCs for a WEC in a parked condition with a fault occurring [3]. Wave loads characterized by $H_{s(5)}$ are required to build DLCs for tidal energy converters in a parked-survival condition under an extreme hydrodynamic load occurring at a peak spring tide.

Perhaps the most challenging analysis for the designer of a maritime structure is characterization of the extreme $n$-year return period environmental conditions, which are then used to build design load cases for evaluating the structural load responses. As the $n$-year return period events are often well beyond the historic period of recorded data, the environmental conditions must be extrapolated from the tails of extreme value distributions, which introduces uncertainty. Best practices, e.g., [1], currently assume historic wave climates are stationary for the extreme value analysis techniques to be valid and provide no methods for adjusting results, e.g., applying a scaling factor. According to [7], the stationary assumption is reasonable for most applications. Trends reported for extreme significant wave heights, particularly large $n$-year return period values, e.g., $n = 100$ years, are subject to large statistical uncertainty [8]. Further, the uncertainties introduced by nonstationary trends are likely less significant than those inherent in the extreme value analysis [9]. Nevertheless, research on historical nonstationary regional trends in wave climates, including mean annual wave heights, e.g., [10], extreme wave heights, e.g., [11], and inter-annual variability [12], suggests significant changes in certain regions. Extreme values of significant wave height are increasing at a higher rate than mean values [12,13]. While the observed trends reported in these studies vary, and there is currently no consensus on the magnitude of these trends, consideration of their implications for design and project risk assessment is warranted.

International standards allow designers the flexibility to choose from a variety of statistical methods to meet minimum design requirements, including simple univariate methods, e.g., peak-over-threshold (POT) methods to calculate the design $n$-year wind, wave or current conditions, or relatively more complex and rigorous bivariate methods, e.g., methods that calculate joint distributions (environmental contours) of statistics characterizing environmental conditions, e.g., the inverse first-order reliability method (IFORM) [14]. A recent review of univariate and bivariate statistical methods commonly used to characterize extreme seas for ocean and coastal engineering applications is given by [7]. These extreme wave statistical methods require the designer to find high-quality data sources of environmental conditions with time series of sufficient duration based on guidance given by international standards, e.g., [15], to accurately calculate return period conditions up to a hundred years or more. Time series of bulk wave parameters like significant wave height, $H_s$, and energy period, $T_e$, for example, can be derived from in-situ measurements at a point or simulated model hindcast databases that provide full coverage of the US coastline at resolutions up to 200 m, e.g., [16].

Muir and El-Shaarawi [17] review data sources and their limitations for calculating $n$-year significant wave height statistics: wave buoys are known to underestimate very large individual waves, e.g., [18]; spatial coverage is limited beyond the point of observation; and measurement durations are typically insufficient to calculate extreme sea state statistics [4]. Wave buoys, however, provide reliable point observations of extreme significant wave height [18], and are the primary data source for ground-truthing wave models.

Spectral wave model hindcasts, using models like WAVEWATCH III®(WWIII) and Simulating WAves Nearshore (SWAN), are an alternative data source for estimating extreme wave height statistics,

and are accepted for wave energy resource assessment and ocean engineering design when validated with buoy measurements [1,2]. Model hindcasts also provide a means to address the noted limitations of spatial coverage and measurement duration from buoy observations. Model performance studies have demonstrated that the third-generation spectral wave models WWIII and SWAN can accurately predict a host of wave energy resource statistics, and that their accuracy is significantly improved with grid refinement [19,20]. Further, these models can examine the spatial variation of wave characteristics at fine resolutions over larger regions encompassing different wave climates, e.g., [20,21]. These models, however, underpredict large waves and extreme *n*-year significant wave heights. The main source of this underbias is due to limitations of most wind reanalysis datasets, namely their inability to resolve fine-scale high-energy wind gusts, e.g., [22,23]. Model performance studies commonly emphasize predictions of common sea state statistics (bulk parameters) on average, e.g., $H_s$ and peak period, $T_p$. Several studies, however, have evaluated model performance predicting extreme significant wave heights at return periods equal to or greater than 1-year, e.g., [7,19,24,25]; and some have examined the use of simple linear correction methods to model-derived estimates of 100-year significant wave heights, $H_{s(100)}$, to reduce model underbias relative to observation-derived estimates [24,25].

Few studies have compared *n*-year return period significant wave heights, $H_{s(n)}$, derived using different univariate extreme value analysis methods, e.g., [9], or compared $H_{s(n)}$ derived using univariate methods with the maximum *n*-year contour $H_s$ values derived using bivariate methods, e.g., [7]. Peak-over-threshold (POT) methods should be used with care and their results should be checked with other methods, e.g., the annual maxima (AM) method, as a threshold value selection can be subjective [1,9]. On the other hand, the AM method, the most common of the so-called *block maxima* methods, can be subject to large uncertainty due to the small sample population [9]. In comparing $H_{s(100)}$ values estimated by univariate analysis (POT method) with maximum values of 100-year environmental contours estimated using bivariate analysis, [7] found values agreed within one meter for twelve of fifteen sites; but no systematic bias was observed.

The present study investigates characterization of extreme wave conditions using different data sources, buoy measurements vs. wave model outputs, and different methods allowed by design standards, univariate methods (AM and POT) and the bivariate environmental contour method. The goal is to understand how estimates of extreme wave characteristics are affected by both factors, and to identify bias trends that may be adjusted. Compared to previous investigations, the present study is much broader in scope in terms of the range of return periods evaluated, the different data sources and methods used, and the number of wave sites and wave climates considered. We compare model skill predicting extreme significant wave heights with 1, 5 and 50-year recurrence intervals, $H_{s(1)}$, $H_{s(5)}$, and $H_{s(50)}$, at approximately two dozen buoy locations spanning several regional wave climates along the US East and West Coasts. We also investigate the use of simple linear corrections to model-derived extreme wave heights to improve their agreement with extreme wave heights derived from buoy measurements. While this model performance evaluation does not consider spectral details that can identify sources of model deficiencies, e.g., [26], it does provide an initial assessment of model performance trends, including bias, and how performance may vary among different sites within a regional wave climate and between different wave climates.

## 2. Methods

### 2.1. Study Region and Buoy Observations

The study region, shown in Figure 1, encompasses potential wave energy project sites within the economic exclusion zones (EEZ) along the US East and US West Coasts [27]. Only National Data Buoy Center (NDBC) stations with hourly $H_s$ and $T_e$ time series spanning long deployment periods of at least 12.5 years were selected, to meet the ISO standard for estimating 50-year return period events using the POT method, which recommends periods of record (POR) at a quarter of the desired return period [15]. Missing data records in these time series reduced the POR to 12.21 years at Station 46050

and to 10.62 years at Station 46054 (Figure 1, Table A1). Also, data in these time series were removed to match modelled hindcast time series generated at three-hour intervals. Depths at these stations vary from 30 m (44009) to 4048 m (41002) and are classified as intermediate and deep wave sites with normalized peak frequencies of $f_P > 0.05 \sqrt{g/h}$.

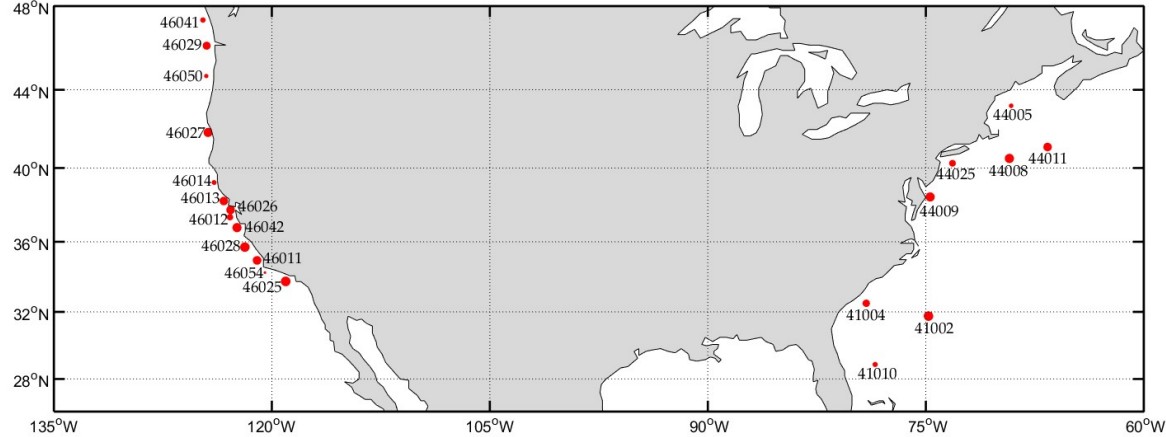

**Figure 1.** NDBC stations along the US East Coast and the US West Coast selected for model performance study. The size of the circle indicates the period of record (POR) with the smallest indicating 11 years (46054), and the largest reflecting 22 years (e.g., 44008).

*2.2. Modelled Hindcast Data Sources*

The present study utilizes modelled hindcast data from several wave model studies to estimate extreme wave statistics: a WWIII 30-year model (v.5.08) hindcast study encompassing all US coastal waters with a spatial resolution of 4 minutes (6–7 km) [28], a high-resolution regional SWAN 32-year model hindcast study for the US West Coast with a spatial resolution of 200–300 m [20], and a high-resolution regional SWAN 32-year model hindcast study for the US East Coast with a spatial resolution of 200 m for coastal areas extending 20 km offshore and a resolution of 1–3 km for the inner-shelf regions [21,29]. At thirty years or more, these model hindcasts exceed minimum requirements for estimating extreme wave heights with return periods up to fifty years. $H_s$ and $T_e$ time series outputs from these model hindcasts at three-hour intervals are used to estimate $H_{s(1)}$, $H_{s(5)}$, and $H_{s(50)}$ and environmental contours $(H_s, T_e)_n$ for comparison with those derived from measured (observation) $(H_s, T_e)$ time series at NDBC stations at eight sites along the East Coast and thirteen sites along the West Coast. For the initial comparison with observation-derived $H_{s(n)}$ estimates, the model-derived $H_s$ time series are limited to the observation POR at each station. These POR-limited model-derived estimates are also used to determine the linear correction, to adjust raw model-derived estimates to better match observation-derived values.

*2.3. Univariate Extreme Value Analysis*

The AM method is applied to estimate $H_{s(5)}$ and $H_{s(50)}$ from $H_s$ time series generated from models and buoy observations for comparison, at three-hour intervals from all data sources at corresponding intervals and spanning the same period of record (POR) to ensure consistency. This method, which fits the yearly maxima $H_s$ to a Gumbel distribution, is an accepted standard detailed in [1]. It is simple to implement and requires no user inputs that can introduce user bias like peak-over-threshold (POT) methods. It does, however, require a minimum POR of approximately 20 years [1]. While this requirement is not met for ten sites, $H_s$ time series for these sites strengthen the correlation between the model- and measurement-derived $H_{s(n)}$ by doubling the population of samples used to derive the linear correction for model-derived $H_{s(n)}$ estimates. As the same POR is used for model and observation data sources, the bias introduced by relaxing this requirement is consistent for both data sources. Following

comparisons of extreme wave height estimates between buoy observations and modelled data sources that are limited to the buoy POR, estimates of extreme wave height using the entire 30 years of the modelled hindcast data sources are also provided herein to examine the dependence of POR length; especially for the buoy stations that did not meet the minimum POR requirements for the AM method.

For low return periods below 5 years, POT methods are recommended [4]. We apply a POT method to estimate $H_{s(1)}$ utilizing the generalized Pareto distribution model (GPD), which has been broadly applied for estimating extreme wave heights, e.g., [7,24,30,31]. For a Type I GPD model, the shape parameter is zero [30], and the cumulative distribution function is

$$F(x) = 1 - exp\left[-\frac{x - \mu}{\sigma}\right] \tag{1}$$

where $x$ is the extreme significant wave height associated with an individual storm, $\mu$ is the threshold significant wave height used to filter the sample population of significant wave heights in the time series, and $\sigma$, the scaling parameter, is the mean value of the excess $(x - \mu)$.

As with other studies, the threshold is selected to be low enough to maintain a minimum population of samples to ensure a robust model fit that limits variance, while high enough to be characterized as a tail sample by the GPD model. Quantile–quantile plots are used to identify the threshold that provides the best-modelled distribution fit [30]. The Wald–Wolfowitz Runs (WWR) test is used to check that samples are independent [32]. The most notable disadvantage of POT method is its sensitivity to the threshold value selected. Despite best efforts to objectively select threshold values, it is difficult not to introduce user bias. Threshold values selected in the present study were 99-percentile $H_s$ values on average, with a standard deviation of several tenths of a meter. For comparison, [7] found that 98-percentile values provided the best regional model fits along the Canadian Pacific Coast when using a Type II GPD model. Vanem [9], in comparing estimates of $H_{s(100)}$ and $H_{s(20)}$ using block maxima and POT methods at a wave site in the North Atlantic, selected 99.5 and 99.95-percentile values for his study to examine the sensitivity of extreme significant wave height to the threshold value.

*2.4. Linear Correction of Modelled Extreme Wave Heights*

A simple linear correction method proposed by Stephens and Gorman [25] is applied to improve the agreement between $H_{s(n)}$ values derived from modelled and measured (observation) data sources at the observation-limited POR, where modelled $H_{s(n)}$ values are scaled by the average relative bias among the study sites,

$$s = \frac{1}{N} \sum_{i=1}^{N} \frac{Mod. \, H_{s(n), \, i} - Obs. \, H_{s(n),i}}{Mod. \, H_{s(n), \, i}} \tag{2}$$

where $N$ is the number of sites, and $i$ is the site index. This correction scaling constant, $s$, should not be confused with the average mean bias, $B$, which considers bias relative to the measurement-derived value to evaluate the model performance.

To compare the goodness of fit between the model and measurement-derived $H_{s(n)}$ values, a linear regression best-fit line is plotted along with the line of equivalence, and $r^2$, and slope ($m$) values are reported (Figures 2–4). The mean absolute relative bias ($B$) is calculated as a summary statistic to quantify this comparison,

$$B = \frac{1}{N} \sum_{i=1}^{N} \left| \frac{Mod. \, H_{s(n), \, i} - Obs. \, H_{s(n),i}}{Obs. \, H_{s(n), \, i}} \right| \tag{3}$$

Here, the bias is relative to the measured data source (observation) rather than the modelled data source as calculated for the scaling factor using (2).

## 2.5. Bivariate Extreme Value Analysis (Environmental Contours)

Environmental contours for 1-, 5- and 50-year sea states within the $(H_s, T_e)$ parameter space are created using an inverse first-order reliability method (IFORM) with principal components analysis (PCA) and inverse Gaussian and normal distribution models as described in [14]. This method is similar to the traditional IFORM summarized in [1]. PCA is applied prior to the IFORM to generate an uncorrelated representation of $(H_s, T_e)$ in terms of principal components $(C_1, C_2)$ that result in better fitting contours. The probability density function of the principal component $C_1$ is parameterized with an inverse Gaussian (Wald) distribution model

$$f(x) = \sqrt{\frac{\lambda}{2\pi x^3}} exp\left[-\frac{\lambda(x-\mu)^2}{2\mu^2 x}\right] \tag{4}$$

with parameters $\mu$ and $\lambda$ determined by maximum likelihood estimation (MLE). Values of $C_2$ are binned based on their corresponding $C_1$ values. Probability density functions of $C_2$ conditioned on $C_1$ are parameterized with a normal (Gaussian) distribution model

$$f(x) = \frac{1}{\sigma\sqrt{2\pi}} exp\left[-\frac{1}{2}\left(\frac{x-\mu}{\sigma}\right)^2\right] \tag{5}$$

with estimates of the $C_2$ normal distribution parameters $\mu$ and $\sigma$ as a function of $C_1$ for each bin modelled with the fitting functions

$$f_\mu(C_1) = \beta_0 + \beta_1 C_1 \tag{6}$$

$$f_\sigma(C_1) = \gamma_0 + \gamma_1 C_1 + \gamma_2 C_1^2 \tag{7}$$

with linear coefficients $\beta_0$, $\beta_1$, and quadratic coefficients $\gamma_0$, $\gamma_1$, $\gamma_2$, determined by a least square regression procedure. The fitted inverse Gaussian distribution model for $C_1$ and fitted normal distributions models for $C_2$, as a function of the mean value of $C_1$ for each bin are used to construct an environmental contour in principal component space for a given return period using the IFORM, which is then transformed back to the $(H_s, T_e)$ parameter space using the equations

$$H_{s_i} = \frac{C_{1_i} v_{1,1} + \left(C_{2_i} - s\right) v_{1,2}}{v_{1,1}^2 + v_{1,2}^2} \tag{8}$$

$$T_{e_i} = \frac{C_{1_i} v_{1,2} + \left(C_{2_i} - s\right) v_{1,1}}{v_{1,1}^2 + v_{1,2}^2} \tag{9}$$

where $v_{1,1}$, $v_{1,2}$, $v_{2,1}$, and $v_{2,2}$ are the PCA rotation coefficients and $s$ is the required adjustment applied to the principal component, $C_{2_i}$, to ensure it is greater than zero in the principal components space. When negative values of $H_{s_i}$ occur at very low values, an artifact of the PCA not being constrained by limitations of the wave physics, they are set to zero. As these sea state occurrences delineate very low wave heights at the base of the contour line, they are not significant for characterizing extreme wave loads.

## 3. Results

### 3.1. Comparison of Univariate Methods

Estimates of $H_{s(n)}$ derived using both AM and POT methods are compared at all study buoy stations, and for all three data sources, in scatter plots shown in Figure 2. Again, for this comparison, the periods of record of model-derived data sources were reduced to match those of the observation-derived data sources for each buoy station—not the entire 30-year POR. The perfect agreement line and slope

and $r^2$ values of linear regression fits are shown to quantify how well estimates are correlated using AM and POT methods applied to the same data source.

The comparison of $H_{s(1)}$ estimates derived using the AM and POT methods, Figure 2a, is presented to illustrate the problem that arises when using the AM method for low return period events, and why it is not recommended practice [4]. The present study adopts the POT method for estimating $H_{s(1)}$ in subsequent analyses. $H_{s(1)}$ estimates derived using the AM method are significantly lower than those derived using the POT method, with all points falling below the line of equivalence. As indicated by the low values for regression slope, this underbias increases with the $H_{s(1)}$ magnitude. Low $r^2$ values indicate large scatter for all data sources, i.e., large and frequent discrepancies between estimates when using these two different methods. By comparison, Figure 2b,c shows that $H_{s(5)}$ and $H_{s(50)}$ estimates are in much better agreement. Points are nearly equally distributed about the line of equivalence, slopes are close to 1 and high $r^2$ values show significant correlations between results using these two different methods applied to the same data source. Agreement is relatively better for $H_{s(5)}$ estimates, but the good agreement for both $H_{s(5)}$, and $H_{s(50)}$ support using the AM method for the present study due to its simpler implementation. Nevertheless, while discrepancies between estimates using the AM and POT method are random, they can vary significantly, similar to [9]. The magnitude and frequency of these large discrepancies increase with return period, but are less sensitive to the POR or data source. In some cases, $H_{s(50)}$ estimates are in perfect agreement, while, in other cases, $H_{s(50)}$ estimates differ by more than 2 m.

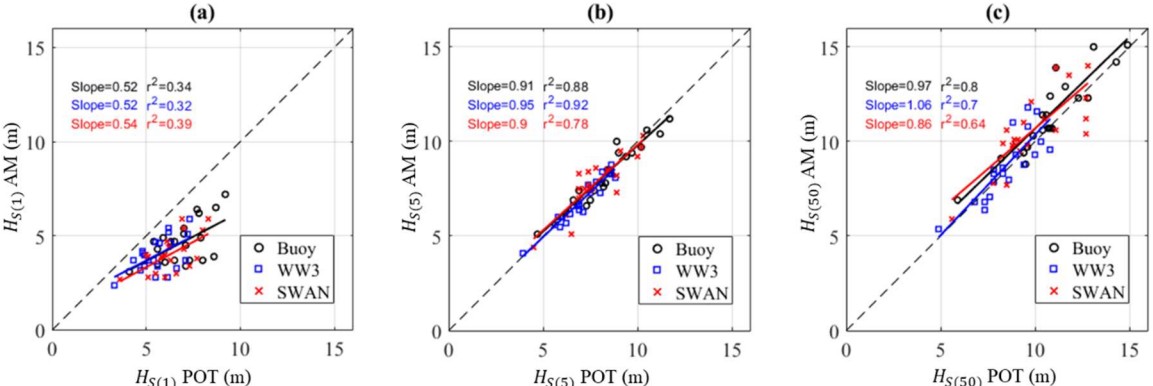

**Figure 2.** Scatter diagrams comparing $H_{s(n)}$ estimates at all buoy stations using the annual maxima (AM) and peak-over-threshold (POT) methods applied to the same data source (buoy observations, WWIII hindcast predictions, SWAN hindcast predictions): (**a**) $H_{s(1)}$; (**b**) $H_{s(5)}$; (**c**) $H_{s(50)}$. Note that, for these comparisons, the periods of record of model-derived data sources were reduced to match those of the observation-derived data sources for each buoy station given in Tables A1–A3.

### 3.2. Comparison of Data Sources for Univariate Analysis

Comparisons between model- and observation-derived $H_{s(1)}$ values (POT Method), and $H_{s(5)}$ and $H_{s(50)}$ values (AM Method) are visualized in scatter plots in Figure 3a–c and summarized in Tables A1–A3. These tables summarize the raw and corrected $H_{s(n)}$ derived by model hindcasts for each site with the percent relative bias in parentheses. Again, for this comparison, the periods of record of model-derived data sources were reduced to match those of the observation-derived data sources for each buoy station—not the entire 30-year POR. Model-derived $H_{s(1)}$, $H_{s(5)}$, and $H_{s(50)}$ values based on the full "30-year" record of the modelled hindcasts, 30 years for WWIII and 32 years for SWAN, are provided in Tables A1–A3 to examine the sensitivity of the model-derived estimates to POR.

With the exception of $H_{s(n)}$, derived from SWAN hindcasts at Stations 46026 and 46054, and $H_{s(50)}$, derived from SWAN hindcasts at Station 41004, the raw values of $H_{s(n)}$ derived using both models, at the observation limited POR, are lower than those derived from buoy observations, as shown in Figure 3a–c and Tables A1–A3. This underbias for model-derived $H_{s(n)}$ is not sensitive

to return period based on the slopes of the linear regression fits. For WWIII, the regression slope $m = 0.78$ for model-derived $H_{s(1)}$, 0.74 for $H_{s(5)}$, and 0.77 for $H_{s(50)}$. For SWAN, the regression slope $m = 0.79$ for model-derived $H_{s(1)}$, 0.87 for $H_{s(5)}$, and 0.86 for $H_{s(50)}$. Model performance predicting extreme wave heights over a broad range of return periods is clearly improved by increasing model spatial resolution. The average model relative absolute bias for WWIII-derived $H_{s(1)}$, $H_{s(5)}$, and $H_{s(50)}$ estimates is approximately 20%, compared to approximately 10% for those derived by the SWAN model.

Raw model- and observation-derived estimates are generally well-correlated based on $r^2$ values, and these correlations are insensitive to return period. Correlations of linear regression fits, shown in Figures 3, 4 and 5a,c, did not improve by separating the study sites by regional wave climate (East Coast and West Coast). Correlations are significantly better between raw WWIII model-derived estimates ($r^2 \geq 0.91$) and observation-derived estimates than those for the raw SWAN model-derived estimates ($r^2 \geq 0.79$), likely due to the aforementioned SWAN model-derived outliers at Stations 41004, 46026 and 46054.

Linear regressions (not shown) indicate significantly stronger correlations and agreement between model and observation-derived 98-percentile significant wave height estimates, $H_{s(98\%)}$, compared to $H_{s(1)}$, $H_{s(5)}$, and $H_{s(50)}$: $m = 0.93$, with $r^2 = 0.97$ for WWIII model-derived 98-percentile wave heights, $H_{s(98\%)}$; and $m = 0.98$ for SWAN model-derived $H_{s(98\%)}$. In other words, the models perform quite well, predicting extreme wave heights up to 98-percentile values. It is the extreme wave heights at return periods $n \geq 1$-year for which model underbias is significant. $H_{s(1)}$ are about twice the height of $H_{s(98\%)}$.

As shown in Tables A1–A3, raw model-derived $H_{s(n)}$ estimates using full 30-year POR are generally within 10% of values using the lower buoy station limited POR. However, discrepancies are particularly large, equal to or exceeding 10%, for several stations. These large discrepancies are not generally associated with low periods of record. Raw model-derived $H_{s(1)}$ estimates using the POT method, as expected, are least sensitive to the POR length. Discrepancies are all below 10% with the exception of SWAN model-derived $H_{s(1)}$ estimates at Stations 46026 and 46054. As the return period increases and the AM method is employed, the number of large discrepancies increases. More large discrepancies are observed for SWAN model-derived estimates than WWIII.

*3.3. Linear Correction of Modelled Extreme Wave Heights*

Corrections applied to the model-derived $H_{s(n)}$, with the scaling factors calculated using Equation (2), significantly improve the agreement between model and observation-derived values. The average relative absolute bias for WWIII is reduced to 3%–4%, and that for SWAN to 6%–7%. Comparing scatter plots with raw model-derived estimates, Figure 3a–c, to corrected model estimates, Figure 3d,e, illustrates how this simple correction adjusts values to better align with lines of equivalence. Increases in the slopes ($m$) of the linear regression fits quantify these improvements, increasing $m = 0.74$–0.78 to 0.90–0.97 for WWIII model-derived estimates, and increasing $m = 0.79$–0.82 to 0.88–0.93 for the SWAN model-derived estimates. As expected, discrepancies between model-derived $H_{s(n)}$ estimates using full 30-year POR and those based on the lower limited POR of the buoy station are similar, whether comparing raw or corrected values.

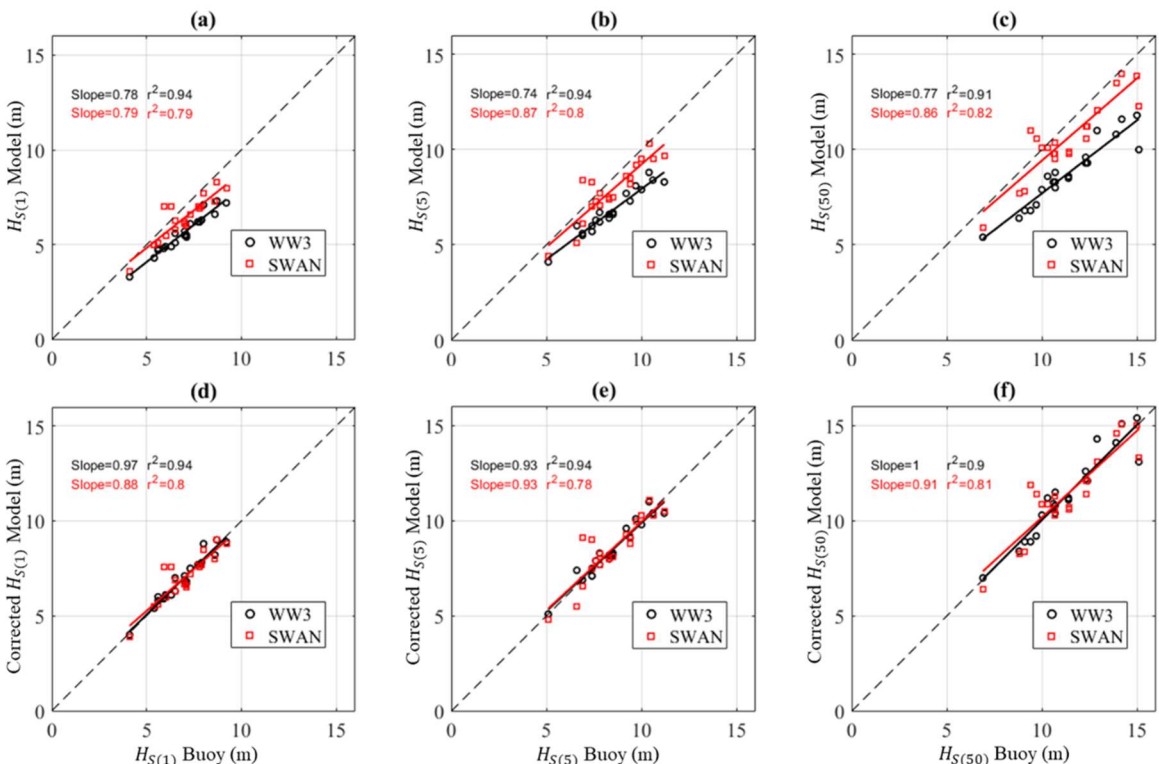

**Figure 3.** Scatter plots at twenty-one NDBC stations comparing $H_{s(1)}$, $H_{s(5)}$, and $H_{s(50)}$ derived from models with those derived from in-situ observations at NDBC stations. Black circles indicate a comparison of WWIII model-derived values with observation-derived values. Red squares indicate a comparison of SWAN model-derived values with observation-derived values. Dashed lines represent lines of equivalence. Solid lines represent lines of best fit from linear regression: (**a**), raw $H_{s(1)}$ models vs. $H_{s(1)}$ buoy (POT method); (**b**), raw $H_{s(5)}$ models vs. $H_{s(5)}$ buoy (AM method); (**c**), raw $H_{s(50)}$ models vs. $H_{s(50)}$ buoy (AM method); (**d**), corrected $H_{s(1)}$ models vs. $H_{s(1)}$ buoy (POT method); (**e**), corrected $H_{s(5)}$ models vs. $H_{s(5)}$ buoy (AM method); (**f**), corrected $H_{s(50)}$ models vs. $H_{s(50)}$ buoy (AM method). Note that raw estimates (**a**–**c**) are adjusted by linear correction with scaling factor, *s*, using (1) to generate corrected estimates (**d**–**f**). Note that, for these comparisons, the periods of record of model-derived data sources were reduced to match those of the observation-derived data sources for each buoy station given in Tables A1–A3.

### 3.4. Environmental Contours

Environmental contours delineating the 1-year, 5-year and 50-year extreme sea states are shown for five selected buoy stations along the East Coast, Figure 4, and five selected stations along the West Coast, Figure 5. These stations were selected to highlight the effect of latitude on extreme sea states for these two US wave climate regions. Each plot compares contours generated using three different data sources: buoy observations, WWIII hindcast data, and SWAN hindcast data. For all stations, these comparisons show that the model-derived contours capture expected trends that affect the contour size as measured by the range of $H_{s(n)}$ and $T_{e(n)}$. As expected, the maximum values of $H_{s(n)}$ increase with return period and with latitude along both coasts, as more energetic wave climates are found in northern latitudes [21,29]. There is no similar north–south trend in maximum values of $T_{e(n)}$.

WWIII model-derived contours are generally smaller than those based on buoy observations, but the WWIII model-based 5-year and 50-year contours at East Coast Station 44025 are nearly identical to those derived from observations. For the West Coast, WWIII and SWAN model-derived contours are similar, but both underpredict the $H_{s(n)}$ and $T_{e(n)}$ range compared to the contour based on observations.

For the East Coast, SWAN model-derived contours are generally in good agreement. Similar to [7], discrepancies are small over a wide range of periods and wave heights for East Coast Station

44005 for 1-year and 5-year contours, and 41004 and 41010 for all *n*-year contours. At the East Coast stations, SWAN model-derived contours exhibit a longer $T_{e(n)}$ range than WWIII contours, and are in good agreement with buoy contours, e.g., Stations 44005, 41004, and 41010 for 1-year and 5-year contours. However, at a few stations, the SWAN contours overestimate the $T_{e(n)}$ range, e.g., 50-year contours at Stations 44005 and 44025.

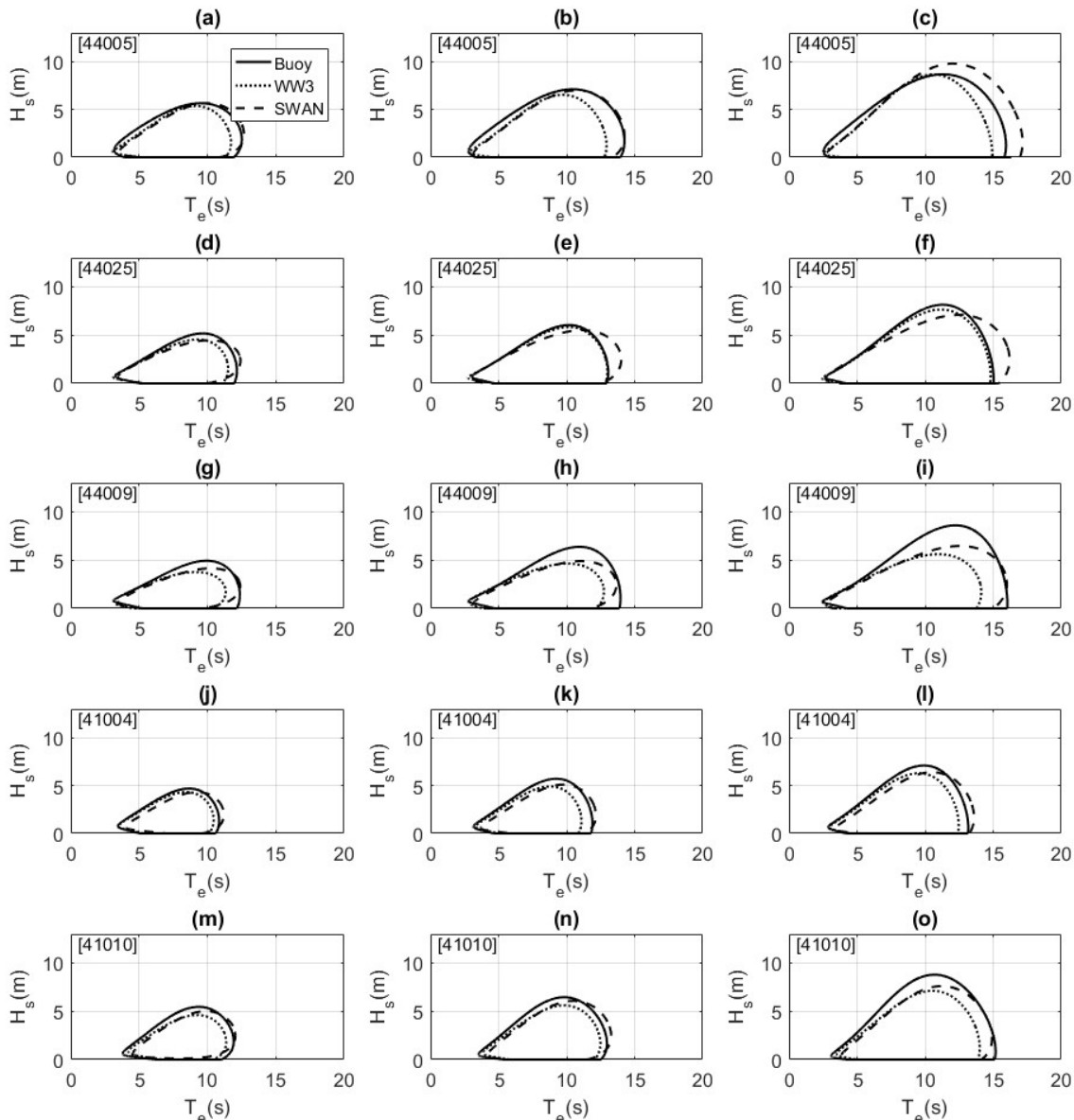

**Figure 4.** Comparison of environmental contours at five selected East Coast buoy stations based on three different data sources: buoy observations (solid line), WWIII (dotted line) and SWAN (dashed line). Each row shows contours at a given buoy station at latitudes in descending order from north to south, and 1-, 5- and 50-year contours shown from left to right: (**a**) 44005, 1-year; (**b**) 44005, 5-year; (**c**) 44005, 50-year; (**d**) 44025, 1-year; (**e**) 44025, 5-year; (**f**) 44025, 50-year; (**g**) 44009, 1-year; (**h**) 44009, 5-year; (**i**) 44009, 50-year; (**j**) 41004, 1-year; (**k**) 41004, 5-year; (**l**) 41004, 50-year; (**m**) 41010, 1-year; (**n**) 41010, 5-year; (**o**) 41010, 50-year.

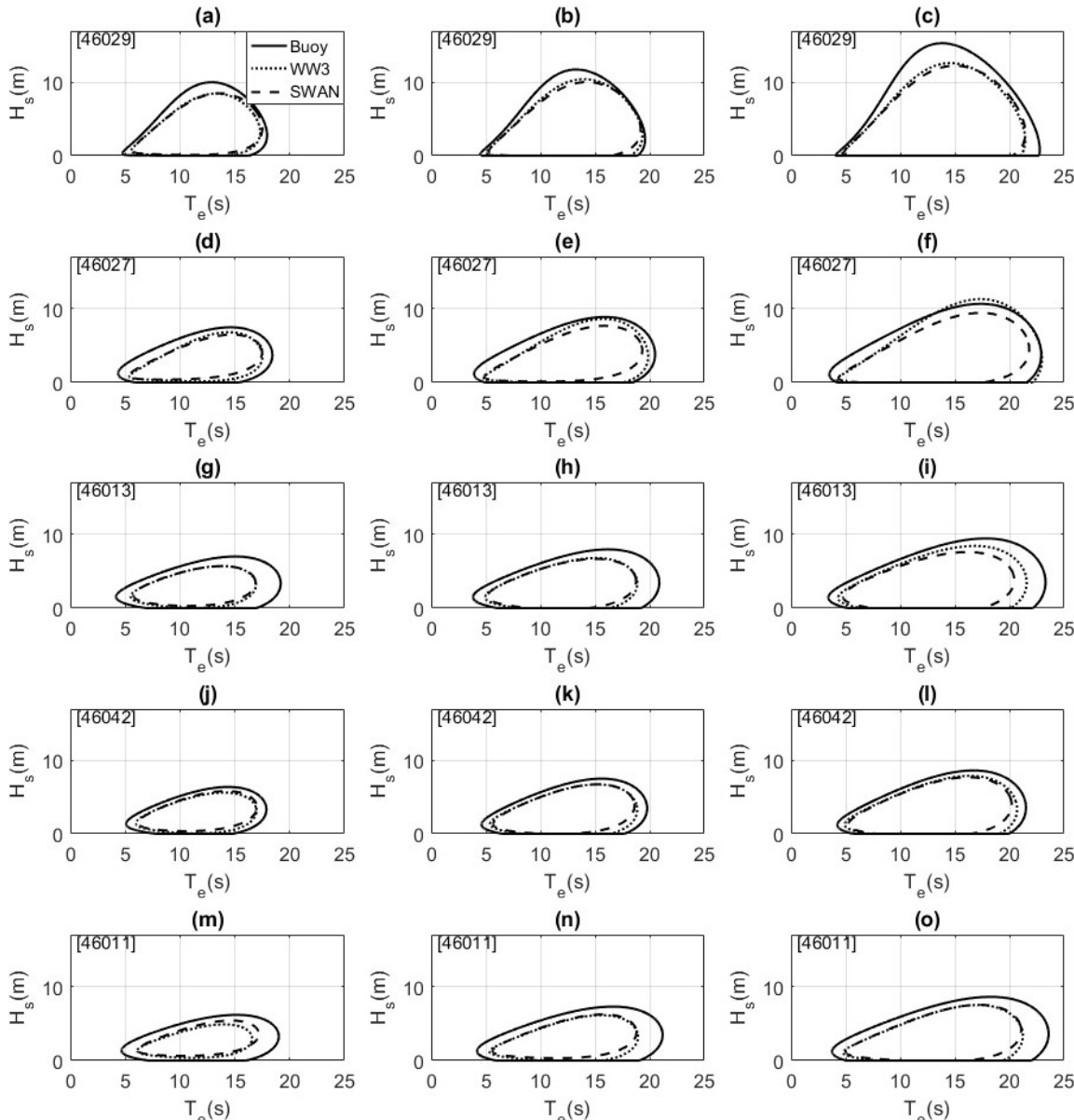

**Figure 5.** Comparison of environmental contours at five selected West Coast buoy stations based on three different data sources: buoy observations (solid line), WWIII (dotted line), and SWAN (dashed line). Each row shows contours at a given buoy station at latitudes in descending order from north to south, and 1-, 5- and 50-year contours shown from left to right: (**a**) 46029, 1-year; (**b**) 46029, 5-year; (**c**) 46029, 50-year; (**d**) 46027, 1-year; (**e**) 46027, 5-year; (**f**) 46027, 50-year; (**g**) 46013, 1-year; (**h**) 46013, 5-year; (**i**) 46013, 50-year; (**j**) 46042, 1-year; (**k**) 46042, 5-year; (**l**) 46042, 50-year; (**m**) 46011, 1-year; (**n**) 46011, 5-year; (**o**) 46011, 50-year.

Maximum values of $H_{s(n)}$ on each contour are compared with values derived using POT and AM methods in Figure 6 for the three different data sources evaluated. Again, results using the AM method are shown to highlight differences at this low return period and why this method should not be used for low return period estimates. As shown in Figure 6a–c, the maximum values of $H_{s(n)}$ on each contour agree well with $H_{s(n)}$ values estimated using the POT method, similar to [7]. Values estimated using the AM method agree less well, likely because the univariate POT methods are more consistent with those used in environmental contour methods; this disagreement is more pronounced for 50-year estimates compared to 5-year, and shows sensitivity to data source at this higher return period. Maximum $H_s$ values on 5-year and 50-year contours exhibit a systematic underbias compared

to estimates of $H_{s(5)}$ and $H_{s(50)}$ based on univariate methods, when using observation and SWAN model hindcast datasources.

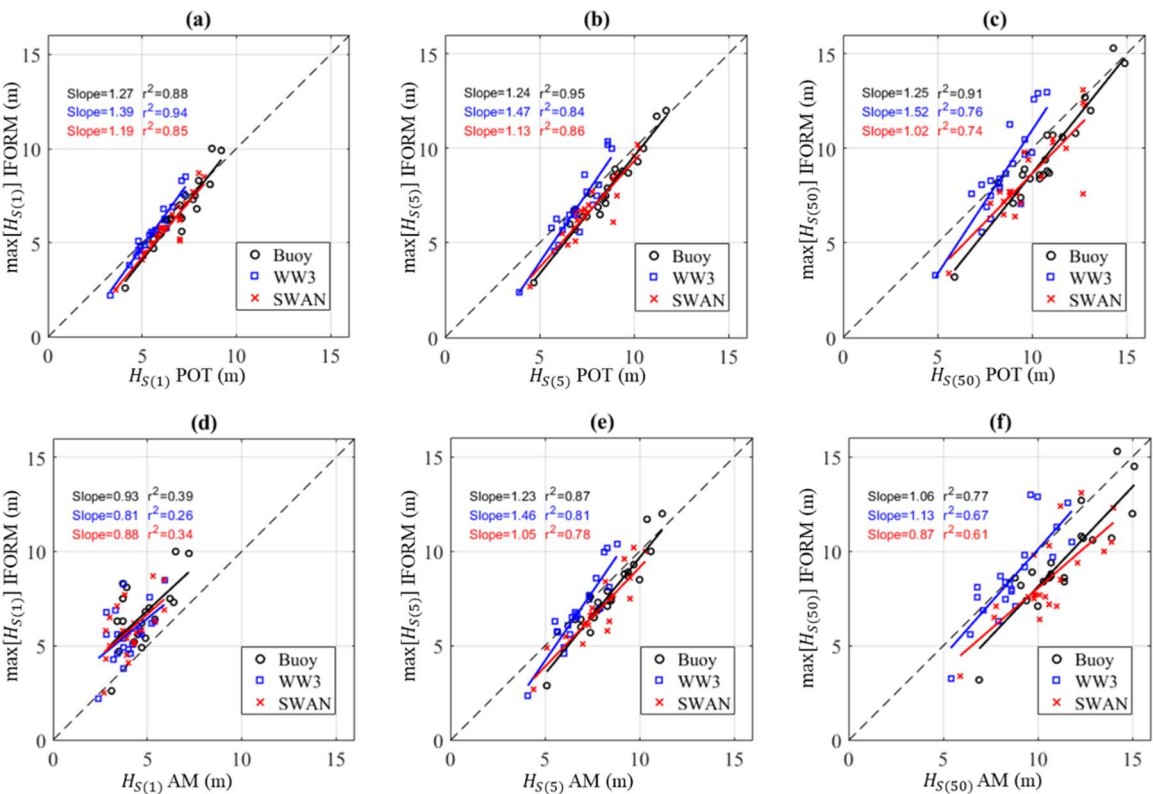

**Figure 6.** Comparison of the maximum values of $H_s$ on each contour with values derived using univariate methods for all buoy stations. Peak-over-threshold (POT) method: (**a**) $H_{s(1)}$; (**b**) $H_{s(5)}$; (**c**) $H_{s(50)}$. Annual maxima (AM) method: (**d**) $H_{s(1)}$; (**e**) $H_{s(5)}$; (**f**) $H_{s(50)}$.

## 4. Discussion

The present study demonstrates that extreme significant wave height estimates using the AM method and POT method can vary significantly, but estimates using the simpler AM method at higher return periods starting at $n$ = 5-years are generally in good agreement with the POT method, even when the minimum 20-year POR requirement is not met [1]. The AM method should, therefore, be considered for estimating higher return period events to avoid the inherent user bias problems of the POT method. The POT method should always be used for low return period events, e.g., $n$ = 1-year [4]. For estimating $n$-year significant wave heights between one and five years, the results of both methods should be compared with a preference for the POT method and good engineering judgment and understanding of the risks of overdesign and underdesign.

Raw model-derived estimates of extreme significant wave height with return periods equal to or greater than 1 year exhibit a systematic underbias compared to observation-derived estimates. This finding, along with similar observations made in previous studies for $H_{s(100)}$ [24,25], indicates that raw model-derived estimates should not be used to characterize extreme wave or sea state conditions with return periods equal to or greater than 1 year for WEC design or project risk assessment. While raw model-derived estimates are still valuable for the relative comparison of project risks between sites, or the spatial distribution of extreme wave conditions, they should be corrected when used to build DLCs for WEC design.

Applications of grid refinement and improved model physics and atmospheric forcing data can significantly reduce model bias in predicting extreme wave heights, including those with high return periods up to 50 years. However, simple linear correction methods applied to model-derived estimates,

$H_{s(n)}$, as demonstrated herein and in other studies [24,25], may offer a more economical approach to offset their observed underbias relative to observation-derived estimates. Without correction, design load cases for extreme sea states based on $H_{s(1)}$, $H_{s(5)}$, and $H_{s(50)}$ could be significantly underestimated [3], and likewise for wave energy project risks based on $H_{s(50)}$ and $H_{s(50)}/H_{s(mean)}$ [6].

Model-derived environmental contours predict similar trends to those derived by buoy observations, but the frequency and magnitude of discrepancies between model- and observation-derived contours is mixed, similar to [7]. In some cases, model-derived contours are in excellent agreement with buoy-derived contours, but generally they underestimate the $H_s$ and $T_e$ range, similar to model-derived estimates using univariate methods. In contrast to [7], results herein indicate a tendency for maximum $H_s$ values on contours to exhibit a systematic underbias compared to estimates based on univariate methods. Unlike univariate methods, there is no discernible improvement in the agreement of model-derived contours with observation-derived contours by increasing model spatial resolution. Further, simple linear correction methods cannot be applied to improve this agreement.

To summarize the main findings of the present study and their implications for design and project risk assessment, consider extreme wave height characterization based on $H_{s(50)}$ for NDBC Stations 46029 amd 46050. The various estimates of $H_{s(50)}$ are given in Table A3. Buoy observations at Station 46029 are limited to only fourteen years at this site, below the 20-year minimum requirement for estimating a 50-year return period event using the AM method [1], but above the 12.5-year minimum requirement for estimating this event using the POT method [15]. The observation-derived estimate of $H_{s(50)} = 14.3$ m using the POT method described herein, therefore, follows the best-practice standard. Using the AM method, which does not follow best practice, $H_{s(50)} = 14.2$ m. Finally, the maximum value of $H_s$ on the 50-year contour generated by bivariate analysis is 15.3 m. In this case, one would typically select $H_{s(50)} = 15.3$ m to characterize the extreme wave condition for WEC design and project risk assessment. It follows best practice and it is a significantly larger and more conservative value than values given using the univariate methods.

For comparison, at Station 46050, with a POR of twelve years, $H_{s(50)} = 14.9$ m using the POT method and $H_{s(50)} = 15.1$ m using the AM method. The maximum value of $H_s$ on the 50-year contour is 14.5 m. In this case, the minimum requirement for the POT method is closely met, and $H_{s(50)} = 14.9$ m would be justified; however, one could also make a case for choosing the more conservative value $H_{s(50)} = 15.1$ m based on the AM method, even though the buoy POR does not satisfy the 20-year minimum requirement. Corrected model-derived estimates base on the full 30-year records, which do satisfy standard requirements for using the AM method, could also be evaluated. For Station 46029, $H_{s(50)} = 13.8$ m, using model-corrected WWIII hindcast data, and $H_{s(50)} = 14.7$ m, using model-corrected SWAN hindcast data. With these results, the largest and most conservative SWAN model-derived estimate $H_{s(50)} = 14.7$ m would be justified if the reduced risk of failure is worth the added cost. For Station 46050, $H_{s(50)} = 14.2$ m using the AM method based on the 30-year model-corrected SWAN hindcast data. As this is less than the observation-derived $H_{s(50)} = 14.9$ m using the POT method, one would typically choose the larger, more conservative value; however, both values satisfy standard practice requirements, and one could alternatively choose the lower value if the added structural design cost is deemed not worth the reduced risk of failure.

## 5. Conclusions

The results presented herein demonstrate several challenges in estimating $n$-year return period extreme significant wave height, $H_{s(n)}$, or sea state, $(H_s, T_e)_n$, to characterize extreme wave conditions for WEC design and project risk assessment. Periods of record for observation-derived estimates typically do not meet the minimum requirement to estimate 50-year return period events, i.e., twelve and a half years for the POT method [15], or twenty years for the AM method [1]. When different methods, e.g., POT and AM, can be used, discrepancies between them are random and can exceed 20%. While validated model hindcasts extend the POR and spatial coverage of data sources, model-derived estimates are systematically underbiased compared to those derived from buoy observations.

Within the scope of the present study, which limits its investigation to coastal waters along the US East and West Coasts, buoy observations that are less than approximately two decades, model hindcast data limited to three decades, and the assumption of stationary wave climate trends inherent in most extreme value analyses, results support the use of simple linear corrections to model-derived estimates, $H_{s(n)}$, as a practical way to extend the POR and spatial coverage of extreme wave statistics, while correcting for underbiased discrepancies compared to observation-derived estimates.

Additional efforts are needed to more rigorously verify these correction methods using a split-sample method, with a population of sea state statistics derived from one set of observations to train the correction scaling, and a population of sea state statistics derived from another set of observations to validate it. Although model underbias is not observed to be affected by differences in regional wave climates, regional effects on scaling factors should not be ruled out and, therefore, should be investigated further. While these simple linear correction methods cannot improve agreement between model- and observation-derived environmental contours, other correction or training techniques, e.g., machine learning, should be investigated to realize similar benefits for bivariate extreme sea state analysis.

Characterization of extreme wave conditions for WEC design and project risk assessment is improved by comparing estimates using different methods and data sources. In this comparison, standard best practices should be followed to the extent possible, but engineering judgment that balances risk of failure and cost must be exercised.

Although the effect of nonstationary wave climate trends on extreme wave conditions is not the focus of the present investigation, it may have significant implications for WEC design and project risk assessment. Simple linear adjustments to extreme wave height estimates, $H_{s(n)}$, based on reported estimates of their projected increases could be applied, but further research is needed to reach consensus on the magnitude and regional distribution of nonstationary trends in extreme wave height.

**Author Contributions:** Conceptualization, V.S.N.; methodology, V.S.N., B.E.S. and S.A.; software, S.A. and B.E.S.; wave model validation, V.S.N., M.N.A., T.W., Z.Y. and R.H.; formal analysis, V.S.N., S.A. and B.E.S.; Investigation, V.S.N. and S.A.; resources, V.S.N. and S.A.; modelled data curation, S.A., T.W. and M.N.A.; writing—original draft preparation, V.S.N.; writing—V.S.N., S.A. and B.E.S.; figures and tables, V.S.N., S.A. and B.E.S.; supervision, V.S.N.; project administration, V.S.N.; funding acquisition, V.S.N. and Z.Y. All authors have read and agreed to the published version of the manuscript.

**Funding:** Sandia National Laboratories is a multi-mission laboratory managed and operated by National Technology and Engineering Solutions of Sandia, LLC., a wholly owned subsidiary of Honeywell International, Inc., for the U.S. Department of Energy's National Nuclear Security Administration under contract DE-NA0003525. This study was also partially funded by the U.S. Department of Energy, Office of Energy Efficiency & Renewable Energy, Water Power Technologies Office under Contract DE-AC05-76RL01830 to Pacific Northwest National Laboratory. This paper describes objective technical results and analysis. Any subjective views or opinions that might be expressed in the paper do not necessarily represent the views of the U.S. Department of Energy or the United States Government.

**Acknowledgments:** The authors thank the reviewers for their thoughtful comments and suggestions, which greatly improved this manuscript; and Arun Chawla and NOAA for providing the WWIII hindcast data used to estimate extreme wave heights and environmental contours.

**Conflicts of Interest:** The authors declare no conflict of Interest.

# Appendix A

**Table A1.** Study sites and comparison of $H_{s(1)}$ using the POT method.

| Station | POR (yrs.) | Dep. (m) | Lat. (°) | Lon. (°) | Buoy | WWIII Raw | WWIII Corr. | SWAN Raw | SWAN Corr. | WWIII Raw | WWIII Corr. | SWAN Raw | SWAN Corr. |
|---|---|---|---|---|---|---|---|---|---|---|---|---|---|
| | | | | | | Station POR | | | | 30-Year POR | | | |
| 41002 | 21 | 4048 | 31.8 | 74.8 | 7.9 | 6.3 (−20%) | 7.8 (−1%) | 7.0 (−11%) | 7.7 (−3%) | 6.3 (0.0%) | 7.9 (1%) | 7.2 (3%) | 8.0 (4%) |
| 41004 | 13 | 39 | 32.5 | 79.1 | 5.6 | 4.7 (−16%) | 5.8 (4%) | 5.1 (−9%) | 5.6 (0.0%) | 4.7 (0.0%) | 5.8 (0.0%) | 5.2 (2%) | 5.7 (2%) |
| 41010 | 13 | 890 | 28.9 | 78.5 | 6.0 | 4.9 (−18%) | 6.1 (2%) | 5.5 (−8%) | 6.0 (0.0%) | 5.3 (8%) | 6.6 (8%) | 6.1 (11%) | 6.7 (12%) |
| 44005 | 22 | 181 | 43.2 | 69.1 | 7.1 | 5.4 (−24%) | 6.7 (−6%) | 6.0 (−15%) | 6.5 (−8%) | 5.7 (6%) | 7.0 (4%) | 6.4 (7%) | 7.0 (8%) |
| 44008 | 22 | 75 | 40.5 | 69.2 | 7.8 | 6.2 (−21%) | 7.7 (−1%) | 6.9 (−12%) | 7.6 (−3%) | 6.5 (5%) | 8.0 (4%) | 7.2 (4%) | 8.0 (5%) |
| 44009 | 19 | 30 | 38.5 | 74.7 | 5.4 | 4.3 (−20%) | 5.4 (0.0%) | 5.0 (−7%) | 5.5 (2%) | 4.6 (7%) | 5.7 (6%) | 5.1 (2%) | 5.6 (2%) |
| 44011 | 20 | 83 | 41.1 | 66.6 | 8.6 | 6.6 (−23%) | 8.2 (−5%) | 7.3 (−15%) | 8.0 (−7%) | 6.9 (5%) | 8.6 (5%) | 7.7 (5%) | 8.5 (6%) |
| 44025 | 15 | 36 | 40.3 | 73.2 | 5.6 | 4.8 (−14%) | 6.0 (7%) | 5.1 (−9%) | 5.6 (0.0%) | 4.9 (2%) | 6.0 (0.0%) | 5.2 (2%) | 5.7 (2%) |
| 46011 | 19 | 465 | 35.0 | 121.0 | 6.5 | 5.1 (−22%) | 6.3 (−3%) | 5.8 (−11%) | 6.3 (−3%) | 5.4 (6%) | 6.7 (6%) | 5.9 (2%) | 6.5 (3%) |
| 46012 | 20 | 209 | 37.4 | 122.9 | 6.5 | 5.6 (−14%) | 7.0 (8%) | 6.3 (−3%) | 6.9 (6%) | 5.9 (5%) | 7.4 (6%) | 6.5 (3%) | 7.2 (4%) |
| 46013 | 21 | 123 | 38.2 | 123.3 | 7.0 | 5.7 (−19%) | 7.1 (1%) | 6.2 (−11%) | 6.8 (−3%) | 6.0 (5%) | 7.4 (4%) | 6.3 (2%) | 7.0 (3%) |
| 46014 | 22 | 356 | 39.2 | 124.0 | 7.7 | 6.2 (−19%) | 7.7 (0.0%) | 7.0 (−9%) | 7.7 (0.0%) | 6.4 (3%) | 7.9 (3%) | 6.9 (−1%) | 7.6 (−1%) |
| 46025 | 21 | 888 | 33.8 | 119.0 | 4.1 | 3.3 (−20%) | 4.0 (−2%) | 3.6 (−12%) | 3.9 (−5%) | 3.4 (3%) | 4.3 (8%) | 3.8 (6%) | 4.2 (8%) |
| 46026 | 21 | 55 | 37.8 | 122.8 | 5.9 | 4.8 (−19%) | 5.9 (0.0%) | 7.0 (19%) | 7.6 (29%) | 5.0 (4%) | 6.2 (5%) | 6.0 (−14%) | 6.6 (−13%) |
| 46027 | 16 | 46 | 41.9 | 124.4 | 7.3 | 6.1 (−16%) | 7.5 (3%) | 6.6 (−10%) | 7.2 (−1%) | 6.4 (5%) | 8.0 (7%) | 6.7 (2%) | 7.4 (3%) |
| 46028 | 18 | 1048 | 35.7 | 121.9 | 7.1 | 5.5 (−23%) | 6.8 (−4%) | 6.0 (−15%) | 6.6 (−7%) | 5.9 (7%) | 7.3 (7%) | 6.3 (5%) | 6.9 (5%) |
| 46029 | 14 | 134 | 46.1 | 124.5 | 8.7 | 7.3 (−16%) | 9.0 (3%) | 8.3 (−5%) | 9.0 (3%) | 7.7 (5%) | 9.5 (6%) | 8.8 (6%) | 9.7 (8%) |
| 46041 | 13 | 128 | 47.4 | 124.7 | 8.0 | 7.1 (−11%) | 8.8 (10%) | 7.7 (−4%) | 8.5 (6%) | 7.4 (4%) | 9.2 (5%) | 8.1 (5%) | 8.9 (5%) |
| 46042 | 17 | 1646 | 36.8 | 122.4 | 7.0 | 5.5 (−21%) | 6.9 (−1%) | 6.1 (−13%) | 6.7 (−4%) | 5.9 (7%) | 7.3 (6%) | 6.4 (5%) | 7.0 (4%) |
| 46050 | 12 | 140 | 44.7 | 124.5 | 9.2 | 7.2 (−22%) | 8.9 (−3%) | 8.0 (−13%) | 8.8 (−4%) | 7.5 (4%) | 9.3 (4%) | 8.7 (9%) | 9.6 (9%) |
| 46054 | 11 | 469 | 34.3 | 120.5 | 6.3 | 4.9 (−22%) | 6.1 (−3%) | 7.0 (11%) | 7.6 (21%) | 5.2 (6%) | 6.5 (7%) | 5.8 (−17%) | 6.4 (−16%) |
| | | | | | *B* | 19.1% | 3.2% | 10.6% | 5.5% | 4.7% | 4.8% | 5.4% | 5.8% |

The values in parentheses for Tables A1–A3 indicate the percent difference between the model hindcast derived value and the buoy measurement derived value. This percent relative difference is calculated as $100 \cdot \left[ \left( Mod.\ H_{s(1)} - Obs.H_{s\ (1)} \right) / Obs.H_{s\ (1)} \right]$. The percent relative difference for 30-year POR is calculated as $100 \cdot \left[ \left( 30y\ POR.\ H_{s(1)} - Sta\ POR.H_{s\ (1)} \right) / Sta\ POR.H_{s\ (1)} \right]$. Note that the mean relative absolute bias, *B*, is the simple average of the absolute values of these relative differences.

**Table A2.** Study sites and comparison of $H_{s(5)}$ using the AM method.

| Station | POR (yrs.) | Dep. (m) | Lat. (°) | Lon. (°) | Buoy | $H_{s(5)}$ (m) Station POR WWIII Raw | | Corr. | | SWAN Raw | | Corr. | | 30-Year POR WWIII Raw | | Corr. | | SWAN Raw | | Corr. | |
|---|---|---|---|---|---|---|---|---|---|---|---|---|---|---|---|---|---|---|---|---|---|
| 41002 | 21 | 4048 | 31.8 | 74.8 | 10.0 | 7.9 | (−21%) | 9.8 | (−2%) | 9.5 | (−5%) | 10.3 | (3%) | 8.2 | (4%) | 10.1 | (3%) | 10.0 | (5%) | 10.8 | (5%) |
| 41004 | 13 | 39 | 32.5 | 79.1 | 7.4 | 6.0 | (−19%) | 7.5 | (1%) | 7.0 | (−5%) | 7.5 | (1%) | 6.2 | (3%) | 7.7 | (3%) | 7.5 | (7%) | 8.1 | (8%) |
| 41010 | 13 | 890 | 28.9 | 78.5 | 7.6 | 6.3 | (−17%) | 7.9 | (4%) | 7.3 | (−4%) | 7.9 | (4%) | 7.2 | (14%) | 8.9 | (13%) | 8.6 | (18%) | 9.3 | (18%) |
| 44005 | 22 | 181 | 43.2 | 69.1 | 8.3 | 6.4 | (−23%) | 8.0 | (−4%) | 7.5 | (−10%) | 8.1 | (−2%) | 6.7 | (5%) | 8.3 | (4%) | 7.7 | (3%) | 8.3 | (2%) |
| 44008 | 22 | 75 | 40.5 | 69.2 | 9.4 | 7.3 | (−22%) | 9.1 | (−3%) | 8.2 | (−13%) | 8.8 | (−6%) | 7.8 | (7%) | 9.7 | (7%) | 8.8 | (7%) | 9.5 | (8%) |
| 44009 | 19 | 30 | 38.5 | 74.7 | 6.6 | 6.0 | (−9%) | 7.4 | (12%) | 5.1 | (−23%) | 5.5 | (−17%) | 5.4 | (−10%) | 6.7 | (−9%) | 6.4 | (25%) | 6.9 | (25%) |
| 44011 | 20 | 83 | 41.1 | 66.6 | 10.6 | 8.4 | (−21%) | 10.4 | (−2%) | 9.5 | (−10%) | 10.3 | (−3%) | 8.0 | (−5%) | 9.9 | (−5%) | 9.3 | (−2%) | 10.0 | (−3%) |
| 44025 | 15 | 36 | 40.3 | 73.2 | 6.9 | 5.6 | (−19%) | 6.9 | (0.0%) | 6.1 | (−12%) | 6.6 | (−4%) | 5.6 | (0.0%) | 7.0 | (1%) | 6.4 | (5%) | 6.9 | (5%) |
| 46011 | 19 | 465 | 35.0 | 121.0 | 7.8 | 6.2 | (−21%) | 7.7 | (−1%) | 7.1 | (−9%) | 7.7 | (−1%) | 6.1 | (−2%) | 7.6 | (−1%) | 6.8 | (−4%) | 7.3 | (−5%) |
| 46012 | 20 | 209 | 37.4 | 122.9 | 7.8 | 6.7 | (−14%) | 8.3 | (6%) | 7.7 | (−1%) | 8.3 | (6%) | 6.9 | (3%) | 8.5 | (2%) | 7.8 | (1%) | 8.4 | (1%) |
| 46013 | 21 | 123 | 38.2 | 123.3 | 8.3 | 6.6 | (−20%) | 8.2 | (−1%) | 7.4 | (−11%) | 8.0 | (−4%) | 6.9 | (5%) | 8.6 | (5%) | 7.7 | (4%) | 8.3 | (4%) |
| 46014 | 22 | 356 | 39.2 | 124.0 | 9.4 | 7.3 | (−22%) | 9.1 | (−3%) | 8.5 | (−10%) | 9.2 | (−2%) | 7.4 | (1%) | 9.2 | (1%) | 8.3 | (−2%) | 9.0 | (−2%) |
| 46025 | 21 | 888 | 33.8 | 119.0 | 5.1 | 4.1 | (−20%) | 5.1 | (0.0%) | 4.4 | (−14%) | 4.8 | (−6%) | 4.0 | (−2%) | 5.0 | (−2%) | 4.5 | (2%) | 4.9 | (2%) |
| 46026 | 21 | 55 | 37.8 | 122.8 | 6.9 | 5.5 | (−20%) | 6.9 | (0.0%) | 8.4 | (22%) | 9.1 | (32%) | 5.6 | (2%) | 7.0 | (1%) | 7.3 | (−13%) | 7.9 | (−13%) |
| 46027 | 16 | 46 | 41.9 | 124.4 | 9.2 | 7.7 | (−16%) | 9.6 | (4%) | 8.6 | (−7%) | 9.3 | (1%) | 7.4 | (−4%) | 9.2 | (−4%) | 8.0 | (−7%) | 8.7 | (−6%) |
| 46028 | 18 | 1048 | 35.7 | 121.9 | 8.5 | 6.6 | (−22%) | 8.2 | (−4%) | 7.5 | (−12%) | 8.1 | (−5%) | 6.7 | (2%) | 8.3 | (1%) | 7.4 | (−1%) | 8.0 | (−1%) |
| 46029 | 14 | 134 | 46.1 | 124.5 | 10.4 | 8.8 | (−15%) | 11.0 | (6%) | 10.3 | (−1%) | 11.1 | (7%) | 8.7 | (−1%) | 10.8 | (−2%) | 10.5 | (2%) | 11.3 | (2%) |
| 46041 | 13 | 128 | 47.4 | 124.7 | 9.7 | 8.1 | (−16%) | 10.1 | (4%) | 9.2 | (−5%) | 10.0 | (3%) | 8.2 | (1%) | 10.2 | (1%) | 9.3 | (1%) | 10.0 | (0.0%) |
| 46042 | 17 | 1646 | 36.8 | 122.4 | 8.5 | 6.7 | (−21%) | 8.3 | (−2%) | 7.5 | (−12%) | 8.1 | (−5%) | 6.8 | (1%) | 8.4 | (1%) | 7.6 | (1%) | 8.2 | (1%) |
| 46050 | 12 | 140 | 44.7 | 124.5 | 11.2 | 8.3 | (−26%) | 10.4 | (−7%) | 9.7 | (−13%) | 10.5 | (−6%) | 8.5 | (2%) | 10.6 | (2%) | 10.3 | (6%) | 11.1 | (6%) |
| 46054 | 11 | 469 | 34.3 | 120.5 | 7.4 | 5.7 | (−23%) | 7.1 | (−4%) | 8.3 | (12%) | 9.0 | (22%) | 6.0 | (5%) | 7.4 | (4%) | 6.6 | (−20%) | 7.1 | (−21%) |
| | | | | | *B* | 19.5% | | 3.4% | | 10.0% | | 6.7% | | 3.8% | | 3.5% | | 6.6% | | 6.6% | |

**Table A3.** Study sites and comparison of $H_{s(50)}$ using the AM method.

| Station | POR (yrs.) | Dep. (m) | Lat. (°) | Lon. (°) | Buoy | Station POR WWIII Raw | Station POR WWIII Corr. | Station POR SWAN Raw | Station POR SWAN Corr. | 30-Year POR WWIII Raw | 30-Year POR WWIII Corr. | 30-Year POR SWAN Raw | 30-Year POR SWAN Corr. |
|---|---|---|---|---|---|---|---|---|---|---|---|---|---|
| 41002 | 21 | 4048 | 31.8 | 74.8 | 13.9 | 10.8 (−22%) | 14.1 (1%) | 13.5 (−3%) | 14.6 (5%) | 10.9 (1%) | 14.3 (1%) | 14.0 (4%) | 15.2 (4%) |
| 41004 | 13 | 39 | 32.5 | 79.1 | 10.0 | 7.9 (−21%) | 10.3 (3%) | 10.1 (1%) | 10.9 (9%) | 8.3 (5%) | 10.9 (6%) | 11.1 (10%) | 12.0 (10%) |
| 41010 | 13 | 890 | 28.9 | 78.5 | 10.7 | 8.8 (−18%) | 11.5 (7%) | 10.4 (−3%) | 11.3 (6%) | 9.8 (11%) | 12.8 (11%) | 12.6 (21%) | 13.6 (20%) |
| 44005 | 22 | 181 | 43.2 | 69.1 | 10.7 | 8.0 (−25%) | 10.4 (−3%) | 9.8 (−8%) | 10.6 (−1%) | 8.4 (5%) | 10.9 (5%) | 9.6 (−2%) | 10.4 (−2%) |
| 44008 | 22 | 75 | 40.5 | 69.2 | 12.3 | 9.3 (−24%) | 12.2 (−1%) | 10.6 (−14%) | 11.4 (−7%) | 10.0 (8%) | 13.1 (7%) | 11.2 (6%) | 12.1 (6%) |
| 44009 | 19 | 30 | 38.5 | 74.7 | 8.8 | 6.4 (−27%) | 8.4 (−5%) | 7.7 (−13%) | 8.3 (−6%) | 6.8 (6%) | 9.0 (7%) | 8.3 (8%) | 8.9 (7%) |
| 44011 | 20 | 83 | 41.1 | 66.6 | 15.0 | 11.8 (−21%) | 15.4 (3%) | 13.9 (−7%) | 15.0 (0.0%) | 9.8 (−17%) | 12.8 (−17%) | 12.0 (−14%) | 12.9 (−14%) |
| 44025 | 15 | 36 | 40.3 | 73.2 | 9.1 | 6.8 (−25%) | 8.9 (−2.2%) | 7.8 (−14%) | 8.4 (−8%) | 6.9 (1.5%) | 9.1 (2%) | 8.2 (5%) | 8.9 (6%) |
| 46011 | 19 | 465 | 35.0 | 121.0 | 10.6 | 8.3 (−22%) | 10.9 (3%) | 9.8 (−8%) | 10.6 (0.0%) | 7.6 (−8%) | 10.0 (−8%) | 8.6 (−12%) | 9.3 (−12%) |
| 46012 | 20 | 209 | 37.4 | 122.9 | 10.3 | 8.6 (−17%) | 11.2 (9%) | 10.1 (−2%) | 10.9 (6%) | 8.6 (0.0%) | 11.3 (1%) | 10.1 (0.0%) | 10.9 (0.0%) |
| 46013 | 21 | 123 | 38.2 | 123.3 | 10.7 | 8.3 (−22%) | 10.8 (1%) | 9.5 (−11%) | 10.3 (−4%) | 8.6 (4%) | 11.3 (5%) | 10.0 (5%) | 10.8 (5%) |
| 46014 | 22 | 356 | 39.2 | 124.0 | 12.4 | 9.3 (−25%) | 12.1 (−2%) | 11.2 (−10%) | 12.1 (−2%) | 9.1 (−2%) | 11.9 (−2%) | 10.6 (−5%) | 11.4 (−6%) |
| 46025 | 21 | 888 | 33.8 | 119.0 | 6.9 | 5.4 (−22%) | 7.0 (1.4%) | 5.9 (−14%) | 6.4 (−7%) | 5.3 (−2%) | 7.0 (0.0%) | 6.0 (2%) | 6.5 (2%) |
| 46026 | 21 | 55 | 37.8 | 122.8 | 9.4 | 6.8 (−28%) | 8.9 (−5.3%) | 11.0 (17%) | 11.9 (27%) | 7.0 (3%) | 9.1 (2%) | 9.6 (−13%) | 10.4 (−13%) |
| 46027 | 16 | 46 | 41.9 | 124.4 | 12.9 | 11.0 (−15%) | 14.3 (11%) | 12.1 (−6%) | 13.1 (2%) | 9.0 (−18%) | 11.8 (−17%) | 10.0 (−17%) | 10.8 (−18%) |
| 46028 | 18 | 1048 | 35.7 | 121.9 | 11.4 | 8.5 (−25%) | 11.1 (−3%) | 9.9 (−13%) | 10.7 (−6%) | 8.2 (−4%) | 10.8 (−3%) | 9.4 (−5%) | 10.1 (−6%) |
| 46029 | 14 | 134 | 46.1 | 124.5 | 14.2 | 11.6 (−18%) | 15.1 (6%) | 14.0 (−1%) | 15.1 (6%) | 10.5 (−9%) | 13.8 (−9%) | 13.6 (−3%) | 14.7 (−3%) |
| 46041 | 13 | 128 | 47.4 | 124.7 | 12.3 | 9.6 (−22%) | 12.6 (2%) | 11.2 (−9%) | 12.1 (−2%) | 9.7 (1%) | 12.7 (1%) | 11.5 (3%) | 12.4 (2%) |
| 46042 | 17 | 1646 | 36.8 | 122.4 | 11.4 | 8.6 (−25%) | 11.2 (−2%) | 9.8 (−14%) | 10.6 (−7%) | 8.5 (−1%) | 11.1 (−1%) | 9.8 (0.0%) | 10.6 (0.0%) |
| 46050 | 12 | 140 | 44.7 | 124.5 | 15.1 | 10.0 (−34%) | 13.1 (−13%) | 12.3 (−19%) | 13.3 (−12%) | 10.4 (4%) | 13.6 (4%) | 13.2 (7%) | 14.2 (7%) |
| 46054 | 11 | 469 | 34.3 | 120.5 | 9.7 | 7.1 (−27%) | 9.2 (−5%) | 10.6 (9%) | 11.4 (18%) | 7.7 (8%) | 10.1 (10%) | 8.4 (−21%) | 9.1 (−20%) |
| | | | | | B | 23.1% | 4.2% | 9.4% | 6.6% | 5.7% | 5.7% | 7.7% | 7.7% |

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
