# Peer review of "Characterization of Extreme Wave Conditions for Wave Energy Converter Design and Project Risk Assessment"

_jmse, doi:10.3390/jmse8040289_

Round 1

Reviewer 1 Report

Dear Authors,

the MS is of interest to the reader, very well written and organised. Results are clearly presented and conclusions fully supported by the study.

I have only very few comments, mostly of style and little suggestions to increase the quality of presentation of your work, that you can find in the attached document.

Author Response

Response Reviewer 1 Comments:

Thank you for the thorough review and constructive criticism. Below is the response to your specific comments.

Line 54: method (IFORM), [5] -> remove the comma. RESOLVED.

Line 67: [9], [10]. -> [9,10]. RESOLVED.

Line 75: wave height statistics: Wave -> wave. RESOLVED.

Line 131: extreme wave statistics: A WWIII 30. -> : a 165; For very low recurrence intervals -> what do you mean by recurrence intervals? Maybe return periods? YES, THIS IS SAME THING, BUT CHANGED TO BE CONSISTENT PER YOUR COMMENT.

Line 167: [24], [18], [25], [6]. -> [24, 18, 25, 6]. RESOLVED.

Line 184:Vanem selected 99.5 and 99.95 -> Vanem is undefined. RESOLVED BY INCLUDING REF#.

Line 406: 4.1. Comparison of univariate methods -> since you discuss also contours (from line 444) I suggest you change or delete this subsection title. DELETED PER YOUR COMMENT.

Line 424: Vanem’s study highlights one of the key disadvantages of the POT method, the…->try to highlight, in the results section, what are the results of your own study, and if they are in agreement with other study, state similarly “as” already observed by; otherwise the results section seems more like a literature review and that’s not good.

Thank you for noticing this issue. Note, that the JMSE journal template instructs the author to make comparison with other studies in the Discussion section. We agree with your assessment and moved this discussion to the methods section to elaborate on the challenge of threshold selection. One of Vanem’s study objectives was to examine the sensitivity of the extreme wave height to the threshold value selected. This was not the objective of the present study, where the POT method selects the threshold to give the best model fit. Further, (1) Vanem only provide extreme wave height estimates for one site, (2) we could not make a direct comparison as we computed Hs(50), not Hs(100) or Hs(20) like Vanem.  Therefore, any comparison with Vanem ends up being vague and more distracting from the discussion.

Line 430: Raw model-derived estimates of extreme significant wave height are systematically biased low -> the language could be revised here. This was replaced with “ … exhibited a systematic underbias.”

Lines 449-457: same comments as before: try to highlight, in the discussion section, what are the results of your own study, and if they are in agreement with other study, state simply “as” already observed by; otherwise the results section seems more like a literature review and that’s not good.

Again, we agree with your assessment. Most of this discussion was moved to the Introduction where past studies (literature) are reviewed. Note that we were following instructions given in the JMSE template provided for the Discussions section, which states, “… authors should discuss the results and how they can be interpreted in perspective of previous studies …” The Results section “… should provide a concise and precise description of the experimental results, their interpretation as well as the experimental conclusions that can be drawn.”

Line 459—474 The conclusion seems a bit weak, when compared to the significant work described in the manuscript. Try to assert more the importance of your work in the conclusion. Highlight more how the linear correction can improve more the extreme sea state estimates instead of referring to [18] and [19]. Moreover: Line 474: However, similar studies, e.g., [18] and [19] suggest that simple linear corrections can be applied broadly to any wave climate if carefully calibrated with buoy observations. -> I don’t understand why you mention this in the conclusion; maybe it is better in the introduction or result section?

Thank you for your input. As with your previous comments regarding misplacement or overemphasis of past studies in Discussions, the Conclusions was modified, deemphasizing findings of similar studies [18, 19].

Just a curiosity: did you started to work already on these machine learning techniques? Not yet, but we hope to start on this soon. Thank you for your interest and feel free to reach out if you would like to collaborate.

Reviewer 2 Report

See attached file

Author Response

Response Reviewer 2 Comments:

Thank you for the thorough review and constructive criticism. Below is the response to your specific comments.

This manuscript considers univariate and bivariate methods for determining extreme sea states and evaluates two spectral wave models, WAVEWATCH III and SWAN, for predicting such states at several ocean buoy locations along the west and east coasts of the US against buoy observations. Simple expressions are provided to correct for underbias in predicted extreme wave heights from the spectral wave model hindcasts. This is particularly useful in application in practice. The authors use state-of-the-art methods to simulate, analyse, and interpret their data. I believe the findings are scientifically correct, and the manuscript is of a very high quality. The manuscript is very readable, and covers a topic of core interest to the readers of JMSE. My recommendation is for the manuscript to be accepted for publication, subject to minor revision.

Three issues for consideration are listed below:

The authors should comment on the spatial convergence tests they performed when selecting the spatial resolutions of the WWIII and SWAN models. Presumably, these are given in the various references [22], [14], [15] and [23] indicated in Section 2.2.

Convergence tests were not performed to select spatial resolutions for the WWIII and SWAN models. Spatial resolutions were specified to conform to best practices given in the International  Electrotechnical Commission (IEC) standards to provide a feasibility level assessment near shore with grid sizes less than 500 meters.

For the case of SWAN model for the East Coast, an unprecedented fine coastal resolution of 200 meters was used, while the 1-3 km inner-shelf resolution is also amongst the finest resolutions implemented for the region.

The coefficients β0, β1, ϒ0, ϒ1 and ϒ2 in equations (6) and (7) should be defined immediately below the equations. This section was revised and these coefficients are now defined.

The authors should comment about the role of large-scale atmospheric anomalies and teleconnections (e.g. the North Atlantic Oscillation NAO, etc.) on the trends in raw buoy data (see e.g. Santo et al., 2016, Decadal variability of extreme wave height representing storm severity in the northeast Atlantic and North Sea since the foundation of the Royal Society, Proc. Roy. Soc., doi.org/10.1098/rspa.2016.0376)

Thank you for your comment. This is a challenging issue to address without distracting from the scope and main thrust of the present study, which is comparing different methods and data sources for extreme value analysis as a designer following best practice guidance given in international standards. 

We added a paragraph in the Introduction section reviewing this issue of nonstationary historic trends in the wave climate and extreme significant wave height, which are observed in many global and regional wave climate studies over the last two decades. While inter-annual trends associated with teleconnections can introduce uncertainty in extreme value analysis, we believe the long term nonstationary historic trends are more important to highlight. The literature on these trends is now quite extensive and beyond the scope of our investigation, but we believe this new paragraph improves the paper by directly addressing the limitation and added uncertainty of extreme value analysis due to nonstationary wave climate trends.   

“Perhaps the most challenging analysis for the designer of a maritime structure is characterization of the extreme n-year return period environmental conditions, which are then used to build design load cases for evaluating the structural load responses. As the n-year return period events are often well beyond the historic period of recorded data, the environmental conditions must be extrapolated from the tails of extreme value distributions, which introduces uncertainty. Best practices, e.g., [1], currently assume historic wave climates are stationary for the extreme value analysis techniques to be valid and provide no methods for adjusting results, e.g., applying a scaling factor. According to [7], the stationary assumption is reasonable for most applications. Trends reported for extreme significant wave heights, particularly large n-year return period values, e.g., n = 100-years, are subject to large statistical uncertainty [8]. Further, the uncertainties introduced by nonstationary trends are likely less significant than those inherent in the extreme value analysis [9]. Nevertheless, research on historical nonstationary regional trends in wave climates, including mean annual wave heights, e.g., [10], and extreme wave heights, e.g., [11], and inter-annual variability [12], suggests significant changes in certain regions. Extreme values of significant wave height are increasing at a higher rate than mean values [12, 13]. While the observed trends reported in these studies vary and there is currently no consensus on the magnitude of these trends, consideration of their implications for design and project risk assessment is warranted.

We also add a paragraph in the Conclusions to highlight the importance of research on nonstationary trends in extreme wave heights:

“Although the effect of nonstationary wave climate trends on extreme wave conditions is not the focus of the present investigation, it may have significant implications for design and project risk assessment. Simple linear adjustments to extreme wave height estimates, , based on reported estimates of their projected increases could be applied; but further research is needed to reach consensus on the magnitude and regional distribution of nonstationary trends in extreme wave height. “

Minor corrections

L44 Change to “… a maritime structure is characterization of the extreme …” CORRECTED

L93 “Fewer model performance studies, …”, L96 “Few studies have …” and L98 “Similarly few studies have …” is somewhat repetitive. Agreed. We have rewritten this section to avoid this redundancy and to provide more clarity.

L156 Change to “… time series for these sites strengthen the …” CORRECTED

L161 “… are also reported to examine …” Give references. These are results that were computed in the present study, not from another study. This was rewritten to state this explicitly for clarity.

L181 Change to “… it is difficult not to introduce …” CORRECTED

L252 Change to “… to illustrate the problem the arises from using the AM CORRECTED
